# A Comparative Study of Growth Performance, Blood Biochemistry, Rumen Fermentation, and Ruminal and Fecal Bacterial Structure between Yaks and Cattle Raised under High Concentrate Feeding Conditions

**DOI:** 10.3390/microorganisms11102399

**Published:** 2023-09-26

**Authors:** Xiaojing Liu, Zhanming Yang, Jinfen Yang, Dongyang Wang, Jianzhang Niu, Binqiang Bai, Wu Sun, Shike Ma, Yanfen Cheng, Lizhuang Hao

**Affiliations:** 1Key Laboratory of Plateau Grazing Animal Nutrition and Feed Science of Qinghai Province, Qinghai Academy of Animal Science and Veterinary Medicine of Qinghai University, Xining 810016, China; 18853819002@163.com (X.L.); 17697415882@163.com (Z.Y.); 15500741942@163.com (J.Y.); 1981990026@qhu.edu.cn (J.N.); binqiangbai@163.com (B.B.); 2019990019@qhu.edu.cn (W.S.); 2002990028@qhu.edu.cn (S.M.); 2Laboratory of Gastrointestinal Microbiology, National Center for International Research on Animal Gut Nutrition, Nanjing Agricultural University, Nanjing 210095, China; wangdy960821@outlook.com; 3State Key Laboratory of Grassland Agro-Ecosystems, Center for Grassland Microbiome, College of Pastoral Agriculture Science and Technology, Lanzhou University, Lanzhou 730000, China

**Keywords:** yaks, cattle, blood metabolism, rumen bacterial, fecal bacterial

## Abstract

This study compared the growth performance, serum biochemical indicators, rumen fermentation parameters, rumen bacterial structure, and fecal bacterial structure of cattle and yaks fed for two months and given a feed containing concentrate of a roughage ratio of 7:3 on a dry matter basis. Compared with cattle, yak showed better growth performance. The serum biochemical results showed that the albumin/globulin ratio in yak serum was significantly higher than that in cattle. Aspartate aminotransferase, indirect bilirubin, creatine kinase, lactate dehydrogenase, and total cholesterol were significantly lower in yaks than in cattle. The rumen pH, acetate to propionate ratio, and acetate were lower in yaks than in cattle, whereas the lactate in yaks was higher than in cattle. There were significant differences in the structure of ruminal as well as fecal bacteria between cattle and yaks. The prediction of rumen bacterial function showed that there was a metabolic difference between cattle and yaks. In general, the metabolic pathway of cattle was mainly riched in a de novo synthesis of nucleotides, whereas that of yaks was mainly riched in the metabolic utilization of nutrients. This study provides a basis for understanding a rumen ecology under the condition of a high concentrate diet.

## 1. Introduction

The grass-withering period is long in the Qinghai–Tibetan plateau due to its harsh environment. Yaks mainly live grazing at the Qinghai–Tibetan plateau. A large number of studies on the adaptability of cattle and yaks show that yaks are better able to adapt to the pasture conditions of the Qinghai–Tibetan Plateau [1,2], and yaks have a higher lignocellulose degradation rate than cattle [3,4]. This is related to the long-term grazing life of yaks, which also leads to a greater abundance of total cellulose-degrading bacteria in the rumen [5].

However, under the condition of long-term low nutrition levels, it is unfavorable for the production performance of yaks; in the cold season, the yaks can lose up to 25% of their body weight [6]. With the increase in the demand for milk and meat products and the gradual transformation of yak breeding modes from initial grazing life to grazing plus supplementary feeding and captive breeding, it is particularly important to study the physiological response of yaks under high-concentrated feeding conditions.

There are many research reports that show that feeding large amounts of a high concentrate diet can improve production performance but cause nutritional metabolic diseases such as rumen acidosis, which changes the structure and composition of the rumen microbial community [7], further leading to rapid fermentation in the rumen and a lower pH due to elevated levels of volatile fatty acids and lactate [8]. In addition, studies report that a large amount of fermentable carbohydrates flowing into the hindgut can cause hindgut acidosis and even affect liver function through the increase in lipopolysaccharide, biogenic amines, and other substances [9,10,11].

However, there is less information on the metabolic capacity, rumen microbiota, and health status of yaks compared to cattle given high concentrate diets. Therefore, in this study, we hypothesized that there is a difference in the metabolic ability of cattle and yaks in response to high concentrate diets, including differences in growth performance, blood biochemistry, rumen fermentation, and ruminal and fecal bacterial structure between yaks and cattle. To achieve this goal, rumen fluid, serum, and fecal samples from yak and cattle after a two-month feeding period were used to investigate differences in relevant indicators.

## 2. Materials and Methods

This experiment was conducted at the Fengju Breeding Professional Cooperative of Datong County, Qinghai Province, China (altitude approximately 2900 m; 37°40′ N latitude and 101°52′ E longitude) in 2021 from July to September. The site has a plateau continental climate. The protocol and all experimental procedures on the animals were approved by the Committee of Animal Use, Academy of Science and Veterinary Medicine, Qinghai University (Protocol QMKY202002).

### 2.1. Feeding and Management

Ten males each of the same age (an approximate age of 36 months) of Datong yaks and native cattle were selected for the study. Before the experiment, the cowshed was thoroughly cleaned and disinfected, and all animals were housed individually in single pens (1.5 m × 3 m). The pre-feeding adaptation period was two weeks, and the data collection period was two months. All the animals were provided a total mixed ration (Table 1) twice daily at 8:00 and 17:00. At the same time, the feeding trough, sink, and cowshed were cleaned, and the experimental animals had free access to feed and water on a round-the-clock basis. Feed and water intake were recorded daily. Animals were weighed before and after the experiment to calculate average daily gain and feed to gain ratio.

### 2.2. Sample Collection

Before morning feeding every week in the formal trial period, the collected dietary samples were mixed evenly and then sampled by a quartering method. The samples were baked to constant weight at 65 °C and then crushed and stored at −20 °C for testing. Basal diet samples were analyzed for DM, ether extract (EE), and CP according to AOAC [12], using the standard procedures: 934.01, 942.05, and 920.39, respectively. NDF and ADF contents were assessed according to Van Soest [13]. The samples of serum and the rumen fluid of all experimental individuals were collected on day 60 before morning feeding. The blood samples were collected via the jugular vein and were then centrifuged at 3200× *g* for 20 min. The serum samples were stored at −80 °C with 1.5 mL sterile microtubes for later analysis. Ruminal fluid samples were collected using a flexible oral stomach tube with a metal strainer. The oral stomach tube was washed with clean water, and the first 200 mL of ruminal fluid was discarded during the collection to avoid contamination between individuals. Collected rumen fluid samples were frozen immediately in liquid nitrogen for the later analysis of bacterial community.

### 2.3. Biochemical Index Determination for the Blood and Fermentation Parameters of Rumen Fluid

Total protein, albumin, globulin, glucose, total cholesterol, triglyceride, aspartate aminotransferase, alanine aminotransferase, alkaline phosphatase, total bilirubin, direct bilirubin, indirect bilirubin, creatine kinase, lactate dehydrogenase, calcium, and phosphorus in the serum were measured by an automatic biochemical analyzer (BS-280, Mindray Bio-Medical Electronics Co., Ltd., Shenzhen, China). The pH value of the rumen fluid was measured immediately after collection with a pH meter (Ecoscan pH 5, Singapore). Lactate was determined with the Lactate Assay Kit (Nanjing Jiancheng Bioengineering Institute, Nanjing, China). For the analyses of the volatile fatty acids, 1 mL of rumen supernatant was added into a 1.5 mL centrifuge tube, and then added to a 200 μL deproteinized metaphosphate solution containing crotonic acid. The mixture was filtered using a filter (size 0.22 µm, SCAA-102; ANPEL, Shanghai, China) after centrifugation at 12,000× *g* for 10 min. Then, the volatile fatty acids were analyzed using GC (Agilent 7890B, Agilent, Palo Alto, CA, USA), keeping a column temperature of 135 °C, an injection temperature of 200 °C, an FID detector temperature of 200 °C, and a carrier gas (N2) pressure of 0.06 MPa [14].

### 2.4. DNA Extraction, 16S rRNA Gene Amplification, and High-Throughput Sequencing for Rumen

The E.Z.N.A^®^ kit (Omega Bio-tek, Norcross, GA, USA) was used to extract the total DNA of rumen bacteria. The concentration and the purity of the extracted DNA were determined using a NanoDrop 2000 UV-vis Spectrophotometer (Thermo Scientific, Wilmington, DE, USA). Qualified genomic DNA samples and corresponding fusion primers were used to configure the PCR reaction system for amplification. The primers were 515-F (5″-GTGCCAGCMGCCGCGGTAA-3″) and 806-R (5″-GGACTACCVGGGTATCTAAT-3″). The PCR amplification products were purified using Agincourt AMPure XP magnetic beads, dissolved in the elution buffer, labeled, and the library was completed. The fragment range and concentration of the library were detected using Agilent 2100 Bioanalyzer. The qualified libraries were sequenced on the HiSeq platform according to the size of the inserted fragments. The reads that matched the primers were intercepted with the software cutadapt v2.6 to remove the contamination of primers and linkers, and the fragments of the target region were obtained. The final clean data were obtained by removing low complexity reads. The sequence was spliced using the software FLASH (Fast Length Adjustment of Short reads, v1.2.11) to obtain the tags of highly variable regions. The sequences with 99% similarity were then assigned to the same operational taxonomic units (OTUs) using the Vsearch plug-in in QIIME2. QIIME 2 software was used to analyze α and β-diversity. The principal coordinate analysis (PCoA) based on the unweighted UniFrac distance was used to compare bacteria communities between yaks and cattle. Based on the LDA value with a threshold of 3, the linear discriminant analysis (LDA) effect size (LEfSe) analysis between groups from 16S rRNA sequencing was analyzed. PICRUSt software was used for predicting the function of the rumen bacteria with the KEGG database [15], and then STAMP was used to analyze the differences between groups. The raw sequence data were subjected to the Sequence Read Archive (SRA) of the NCBI under accession number PRJNA931567.

### 2.5. Statistical Analysis

The analysis of blood biochemical indexes was conducted by an independent sample *t*-test. The *p*-value < 0.05 was considered significant, and the results were presented in the form of mean ± standard error of the mean (SEM) (SPSS 21 version, IBM Corp., Chicago, IL, USA). The drawing of different indicators was conducted using GraphPad Prism 7.0 (GraphPad software, San Diego, CA, USA). The analysis of rumen and fecal bacteria were plotted using the online website (https://www.omicstudio.cn; https://www.bioincloud.tech), and accessed on 1 April 2022.

## 3. Result

### 3.1. Growth Performance, Blood Biochemical Indices, and Rumen Fermentation Parameters

At the same age, yaks had lower dry matter intake, daily water intake, average daily gain, and feed to gain ratio than those of cattle (*p* < 0.050; Table 2). The albumin/globulin ratio, urea, and uric acid were significantly higher in yaks than in cattle (*p* < 0.050; Table 3). Aspartate aminotransferase (*p* = 0.012), indirect bilirubin (*p* = 0.021), creatine kinase (*p* = 0.047), lactate dehydrogenase (*p* < 0.001), and total cholesterol (*p* = 0.001) were significantly lower in yaks than that in cattle. No differences were found in the serum concentrations of alanine aminotransferase, total protein, albumin, globulin, alkaline phosphatase, total bilirubin, direct bilirubin, glucose, calcium, phosphorus, and triglyceride (*p* > 0.05) in yaks and cattle (Table 3). The acetate, acetate to propionate ratio, and pH of rumen fluid were significantly higher in cattle than in yaks, while the lactate content was significantly higher in yaks than in cattle (*p* < 0.05). No significant differences were found in other rumen fermentation parameters (*p* > 0.05; Table 4).

### 3.2. Ruminal and Fecal Bacteria Structure

Alpha diversity was measured by evenness, observed features, Faith’s PD, and Shannon’s diversity index. Alpha-diversity metrics showed no significant difference between cattle and yaks in fluid samples (*p* > 0.05; Figure 1). Based on the unweighted UniFrac distance, a principal coordinate analysis (PCoA) showed that there was an obvious difference in the rumen bacterial community between yaks and cattle (Figure 2A). The Venn diagram shows that in rumen fluid samples, common OTUs are 1250 between cattle and yaks, and unique OTUs are 790 and 1122 between cattle and yaks, respectively (Figure 2B). Differential bacteria were identified from the kingdom to genus levels using the LEfSe analysis by setting the LDA value to 3. The results showed that the rumen bacterial structure of cattle and yaks was significantly different (Figure 2C). At a phylum level, the main bacteria of cattle and yaks were Firmicutes and Bacteroidetes. The relative abundance of Spirochaeae in the rumen fluid of cattle was significantly higher than that in yak (*p* < 0.05, Figure 3A). At the genus level, the bacteria with differences were Prevotella 1 (*p* = 0.024), Quinella (*p* = 0.026), Unclassified Bacteroides BS11 gut group (*p* = 0.049), Villonellaceae UCG-001 (*p* = 0.008), Bacteroides BS11 gut group (*p* = 0.001), Treponema 2 (*p* = 0.038), Ruminococcus 1 (*p* = 0.008), and Uncultured-Bacteroidales BS11 gut group (*p* = 0.001; Figure 3B).

After binning reads at 99% similarity, a total of 6533 OTUs were detected by 16S rRNA gene sequencing; the sequence numbers in the fecal samples of cattle and yaks were 56,785 ± 891 and 58,054 ± 433, respectively. Alpha-diversity metrics showed no significant difference between cattle and yaks in fecal samples (*p* > 0.05; Figure 4). The principal coordinate analysis (PCoA) showed that there was little difference in the fecal bacterial community between yaks and cattle (Figure 5A). The Venn diagram shows that in fecal samples, common OTUs were 1336 between cattle and yaks, and unique OTUs were 732 and 622, respectively (Figure 5B). LEfSe analysis by setting the LDA value to 3 showed that the fecal bacterial structure of cattle and yaks was significantly different (Figure 5C). At the phylum level, the main bacteria of cattle and yaks were Firmicutes and Bacteroidetes. The relative abundance of Spirochaeae in the fecal samples of cattle was significantly lower than that in yaks (*p* < 0.05, Figure 6A). At the genus level, Treponema 2 (*p* = 0.033), Phascolactobacterium (*p* = 0.007), and Ruminococcaceae (*p* = 0.037) were significantly different in the feces of yaks and cattle (Figure 6B).

In addition, the function prediction results of ruminal bacteria show that the pathway of S-adenosyl-L-methionine cycle I, adenosine deoxyribonucleotides denovo biosynthesis II, guanosine deoxyribonucleotides denovo biosynthesis II, and NAD biosynthesis I (fromaspartate) were significantly rich in the cattle (*p* < 0.05), whereas the pathway of 4-aminobutanoate degradation V, sulfate reduction I (assimilatory), super pathway of menaquinol-8 biosynthesis II, sulfate assimilation, and cysteine biosynthesis were significantly rich in the yaks (*p* < 0.05) (Figure 7). Based on Spearman’s correlation analysis, the different genera of rumen bacteria and phenotypic bacteria are indicators of significant correlation (|r| > 0.6, *p* < 0.05; Figure 8).

## 4. Discussion

### 4.1. Production Efficiency, Serum Biochemistry, and Rumen Fermentation Parameters

Under a high concentrate diet, cattle showed a higher average daily gain, which may be due to the fact that yaks mainly graze, while cattle mainly feed in barns providing high concentrate diet. Accordingly, buffalo also show lower daily weight gain than bovines [16]. However, compared with cattle, yaks had a significantly lower feed to gain ratio, indicating that, given a high concentrate diet, yaks have a better feed utilization capacity than cattle. These results are consistent with other studies [17,18,19] in which yaks have been reported to have efficient energy utilization. A high concentrate diet can cause changes in the nutrient metabolism of the body, hence affecting the immune response and health of the body [20], which can be reflected through blood biochemical indicators. Alkaline phosphatase is an important physiological index that can reflect liver function and protein metabolism [21]; it increases in the serum with disease. In this study, the alkaline phosphatase content of cattle and yaks was between 17.5 to 387 U/L, which is within the normal range [22]. However, the content of alkaline phosphatase in cattle was significantly higher than that of yaks. Whether it had pathological significance or not should be specifically discussed in combination with gene expression and tissue slices. The albumin/globulin ratio represents the synthesis of blood protein, which is widely used in clinical biochemistry to identify abnormal protein synthesis. When it is below the normal range, it may represent inflammation, immune deficiency, or liver and kidney diseases [23]. In a study, the albumin/globulin range was 0.45–1.31 in dairy cows [24]. In this study, the value in the serum of cattle and yak was within the normal range, indicating that there was no obvious metabolic disease in both of them under a high concentrate diet. Urea is the final product of protein, which was negatively correlated with nitrogen use efficiency in ruminants [25]. In this study, the urea content in the serum of cattle was significantly lower than that of yaks, indicating that the higher nitrogen position and protein synthesis of cattle under a high concentrate diet led to a higher average daily weight gain in cattle. There is also research that reports that the serum urea level in yaks was also higher than that found in buffalo [26]. The reference range of creatine kinase in buffalo serum was 14.66–309.80 (U/L) [27]. The creatine kinase of yak was within the normal range, but the content in cattle was significantly higher and was also higher than the normal range. The reason may have been that the differences between breeds caused differences in the normal range. However, it is undeniable that the abnormal rise of creatine kinase in the serum may have been due to the cell damage caused by metabolic disorders, which caused its contents to be released into the serum [28]. As a marker of cell damage and inflammatory mediators, lactate dehydrogenase is often used as an indicator for clinical diagnosis [29]. The reference range of lactate dehydrogenase in buffalo serum was 186.72–917.43 (U/L) [27]. In this study, the level of lactate dehydrogenase in cattle and yak serum was beyond the normal range, and it was significantly higher in cattle than in yak. This may be due to the inflammatory reaction of cells, which led to the abnormal increase in lactate dehydrogenase content. The total cholesterol level in buffalo was 0.90–1.99 mmol/L [27]. In addition, the level of total cholesterol was significantly lower in yaks than in cattle, which also reflected the same level in buffalo [26]. Cholesterol is actively metabolized to produce bile acid [30], and bile acid has a strong function of decomposing adipose tissue and protein, enabling the small intestine to quickly decompose these nutrients and absorb them. Thus, it plays a very important role in the metabolism and absorption of body nutrients [31]. Bile acid can also participate in a variety of signal pathways in the body, including fibrosis and immune homeostasis, to manage the immune and inflammatory responses of the body [32,33]. In this study, the total cholesterol level of cattle was significantly higher than that of yaks and exceeded the normal range, which may have been caused by its cholesterol metabolism disorder.

Both pH and the concentrations of VFAs are the main indicators of the status of rumen fermentation [34]. In this study, the ratio of acetate to propionate ranged between 3.10 and 2.67. A ratio greater than 2.2 was reported to have a positive effect on the rumen and productive performance of cattle [35], which indicates that all the yaks and cattle had a well-functioning rumen. The acetate was significantly higher in cattle than in yaks in the study, which is closely linked to fiber degradation [36]. This shows that cattle are more effective in utilizing fiber in high concentrate diets compared to yaks. In addition, the lactate content was significantly higher in yaks than in cattle; this may also indicate that yaks can more effectively utilize starch in high concentrate diets, which further reduces the pH value in the rumen of yaks. It may be an important reason why the rumen pH of yaks was lower than that of cattle. To sum up, under a high concentrate diet, cattle had higher average daily gain than yaks, but yaks had a healthier metabolism and lower feed to gain ratio. These characteristics may be related to the long-term survival conditions and diet level of cattle and yaks.

### 4.2. Rumen and Fecal Bacterial Structure in Cattle and Yaks

The rumen microbial structure was affected by the proportion of concentrate in the diet [37]. A microbial structure influences the health status of animals. The higher the microbial diversity, the better the health of the animals [38]. The α-diversity index of yaks tends to be higher than that of cattle, which may be the result of a feeding forage with high fiber content around the year, and it contains many rare and precious types of grass, which may help increase the diversity of rumen bacteria in yak and improve immunity. At the genus level, Prevotella_1 was the predominant genus, which has a wide metabolic niche due to its genetic relatedness or high genetic variability, which enables it to occupy different ecological niches within the rumen [39]. This genus utilizes starch and proteins to produce succinate and acetate [40]. Therefore, the difference between acetate and lactate may be partly attributed to Prevotella_1. In this study, the relative abundance of Prevotella in the rumen and level of aspartate aminotransferase in serum of cattle were higher than in yaks. In a previous study [41], Prevotella has been shown to have a role in mediating inflammatory reactions, which results in the degeneration of liver adipose tissue and increased inflammation, consequently increasing the serum aspartate aminotransferase. This may be an important reason why the content of aspartate aminotransferase in cattle serum was significantly higher than that in yaks. However, the specific mechanism of Prevotella-mediated inflammatory response needs further study. It has been reported that Quinella is significantly higher in grazing yaks than in cattle [42], which can be due to the fact that lactate and propionate formation are reportedly the major fermentation pathways of Quinella. An analysis of the Quinella genome showed that there is a potential gene for hydrogen uptake but no metabolic pathway for hydrogen production [43]. In addition, in the correlation analysis results of this study, Quinella and lactate content were significantly positively correlated, which may be an important reason for the high lactate content in the rumen of the yaks. Bacteroidales BS11 has been observed to have a higher relative abundance in moose given a diet mainly composed of hemicellulose and lignin [44]. Genome-enabled metabolic analyses results showed that these bacteria can ferment hemicellulose monomeric sugars to short-chain fatty acids. The relative abundance of this genus in the rumen of the yaks was significantly higher than that of the cattle. Veillonellaceae belongs to Firmicutes, which can degrade and utilize cellulose to provide hydrogen and energy. A high relative abundance of Bacteroidales BS11 and Veillonellaceae in the rumen of yaks may be the result of their adaptation to a forage with high hemicellulose and lignin content for a long time, further causing differences among breeds. Even under a high concentrate diet, yaks still had a higher number of Bacteroidales BS11 and Veillonellaceae than cattle. It has been reported that Treponema is a common potential pathogen, which plays a very important role in mediating inflammatory reactions [45,46]. However, there are few reports on its pathogenicity, which may be because its metabolites can promote the expression of inflammatory factors and lead to the generation of body diseases. Therefore, a higher number of Treponema in the rumen of cattle may aggravate the body’s inflammatory response, indicating how it interacts with the host during the feeding of high concentrate diets, and it plays an important role in mitigating the damage caused by high concentrate diets.

In addition, the prediction results of the rumen bacterial function showed that there was metabolic difference potential between cattle and yaks. In general, the metabolic pathway of cattle was mainly oriented towards the de novo synthesis of nucleotides, while that of yaks was mainly oriented towards the metabolic utilization of nutrients. The 4-aminobutanoate is a major endogenous neurotransmitter in the mammalian brain, which can generate succinic semialdehyde in the rumen under the action of aminobutyric acid transaminase and convert it into succinate, entering the tricarboxylic acid cycle. The prediction result of the rumen bacterial function also shows that yaks have a more significant enrichment of 4-aminobutyrate degradation pathway than cattle, which means that yaks have more energy substances to produce [47]. Sulfate in the diet is biochemically reduced to sulfide by sulfate-reducing bacteria in the rumen, which can compete with the methane generation pathway to reduce methane production [48]. It is reported that compared with a high-fiber diet, a high-starch diet significantly enhances rumen fermentation abilities, reduces methane production by inhibiting the proliferation of methane bacteria, and promotes the proliferation of hydrogen trophic bacteria with respiratory function. In this study, the prediction results of rumen bacterial function show that yaks had a stronger sulfate reduction capacity than cattle, so it can be assumed that under a high concentrate diet, yaks have a stronger ability to inhibit methane generation than cattle [49]. This is consistent with the comparative differences of rumen bacterial such as Quinella in yaks and cattle. Cysteine is the end product of the sulfate assimilation pathway, which widely exists in organisms [50]. It is the starting material for the synthesis of methionine, and also participates in the formation of biotin, thiamine, etc. Menaquinol-8 is considered a vitamin that plays a very important role in promoting the absorption of proteins and lipids [51]. Therefore, it is speculated that the higher levels of menaquinol-8 biosynthesis and sulfate assimilation and cysteine biosynthesis may have resulted in more nutrient metabolization and absorption under a concentrate diet in yaks than in cattle.

The results of the fecal bacterial diversity analysis showed that Ruminococcaceae play an important role in the degradation of cellulose and fermentation of plant fibers [52] and are associated with the feed efficiency of dairy cattle [53]. Treponema 2 is a fibrolytic bacteria belonging to Spirochaetae, which can digest hemicellulose. These bacteria have been shown to increase in relative abundance in the rumen of sheep fed a high fiber diet [54]. The results of this study show that the number of Spirochaetae and Treponema 2 were significantly higher in yaks than in cattle. Therefore, it is speculated that the fermentation ability of the hindgut of yaks may be better than that of cattle. However, a specific comparison of the fermentation capacity of the hindgut of yaks and cattle should be discussed in combination with specific research on the disappearance rate of nutrients in the hindgut and fermentation parameters. In this study, whether α-diversity or β-diversity, the differences between the rumen bacterial structure of cattle and yaks are more obvious than that of their feces. A possible reason is that the rumen is the main place for ruminants to digest nutrients. Most nutrients in the diet are digested in the rumen, and only a small amount of undigested nutrients enter the hindgut. The diet is the most important factor affecting intestinal microorganisms; therefore, the difference of rumen microorganisms between cattle and yak is greater than fecal microorganisms. However, at the phylum level, the Spirochaetaes between cattle and yaks show differences not only in rumen, but also in feces, which plays an important role in degrading plant polymers materials such as xylan, pectin, and arabinogalactan [55]. Interestingly, the relative abundance of Spirochaetae in cattle is higher than that of yaks in the rumen, but it is the opposite in the feces, which may be due to the difference of available substrates. Most of the plant polymer materials in cattle are fermented in the rumen, while the unused polymer materials in yaks flow into the hindgut for fermentation. Therefore, it is speculated that the digestive capacity of the hindgut of yaks may be stronger than that of cattle; however, further discussion is required in combination with specific research. It has been reported that the characteristics of the rumen and fecal microbiome of cattle on an alfalfa diet differ slightly between the fecal microbiota of bloated versus non-bloated steers compared with rumen microbiota [56]. It is possible that in the same host, the physiological functions of microorganisms colonized in different parts are different, which is consistent with the research of de Oliveira [57].

## 5. Conclusions

In general, yaks and cattle have mainly grazed and been bred in captivity for a long time, respectively. When fed a high concentrate diet, they showed a difference in growth performance, blood biochemical indices, rumen fermentation parameters, and rumen microbiota. In addition, the fecal bacterial structures of cattle and yaks are also different, suggesting that yaks may have a stronger digestive capacity in the hindgut.

## Figures and Tables

**Figure 1 microorganisms-11-02399-f001:**
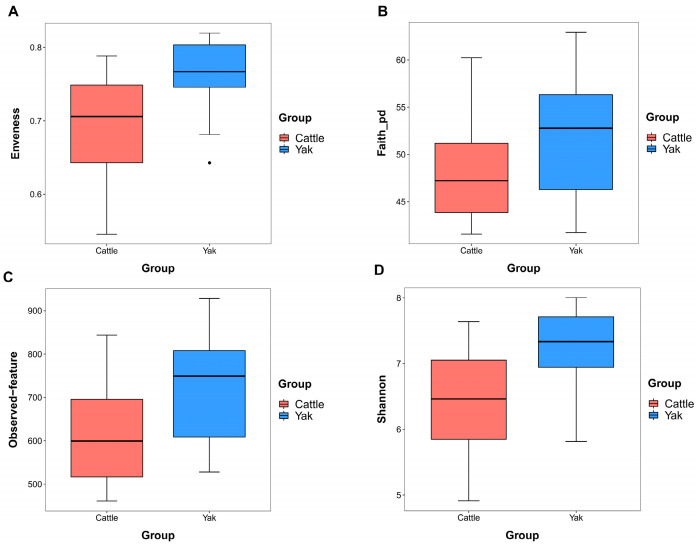
α diversity and richness in ruminal bacteria in yaks and cattle given high concentrate diets. Alpha diversity indices, including Enveness (**A**), Faith_pd (**B**), Observed feature (**C**) and Shannon (**D**). The black dots in Figure (**A**) represent outliers.

**Figure 2 microorganisms-11-02399-f002:**
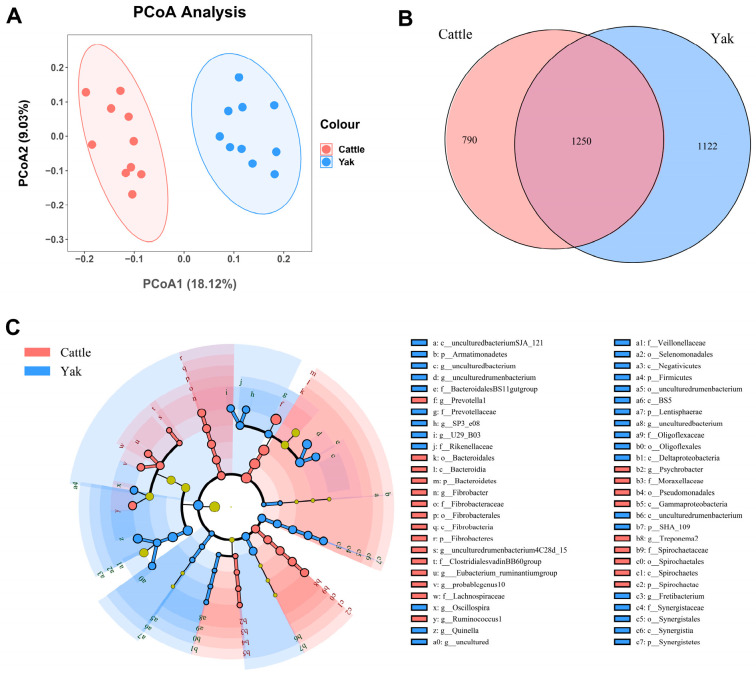
(**A**) PCoA plots and (**B**) Venn diagrams of the rumen bacterial community. (**C**) a cladogram shows rumen bacteria from the phylum level to genus level based on the linear discriminant analysis (LDA) effect size (LEfSe) analysis in yaks and cattle given high concentrate diets.

**Figure 3 microorganisms-11-02399-f003:**
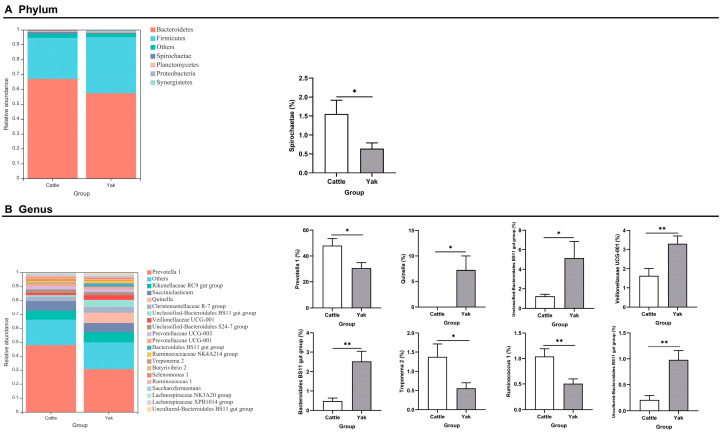
(**A**) The rumen bacterial composition at the phylum level (the relative abundance > 0.5%) and bacteria in cattle and yaks. (**B**) The rumen bacterial composition at the genus level (top 20) and bacteria in cattle and yaks given high concentrate diets. A single asterisk (*) indicates a significant difference at *p* < 0.05, while double asterisks (**) indicate a significant difference at *p* < 0.01.

**Figure 4 microorganisms-11-02399-f004:**
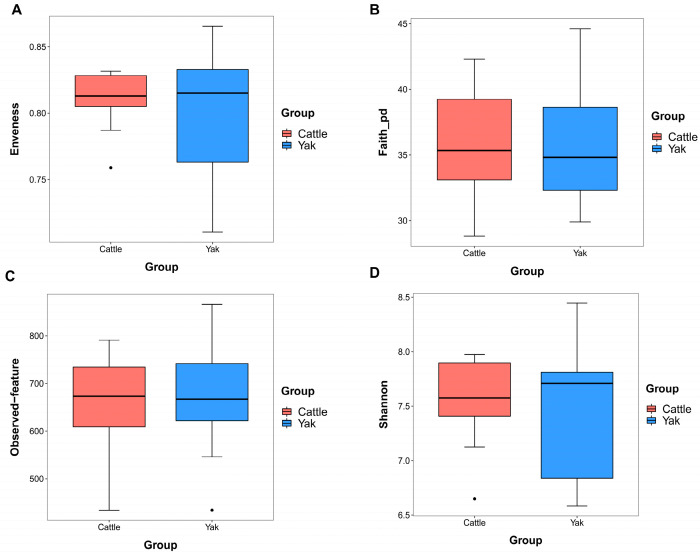
α diversity and richness in fecal bacteria in yaks and cattle given high concentrate diets. Alpha diversity indices, including Enveness (**A**), Faith_pd (**B**), Observed feature (**C**) and Shannon (**D**). The black dots in Figure (**A**,**C**,**D**) represent outliers.

**Figure 5 microorganisms-11-02399-f005:**
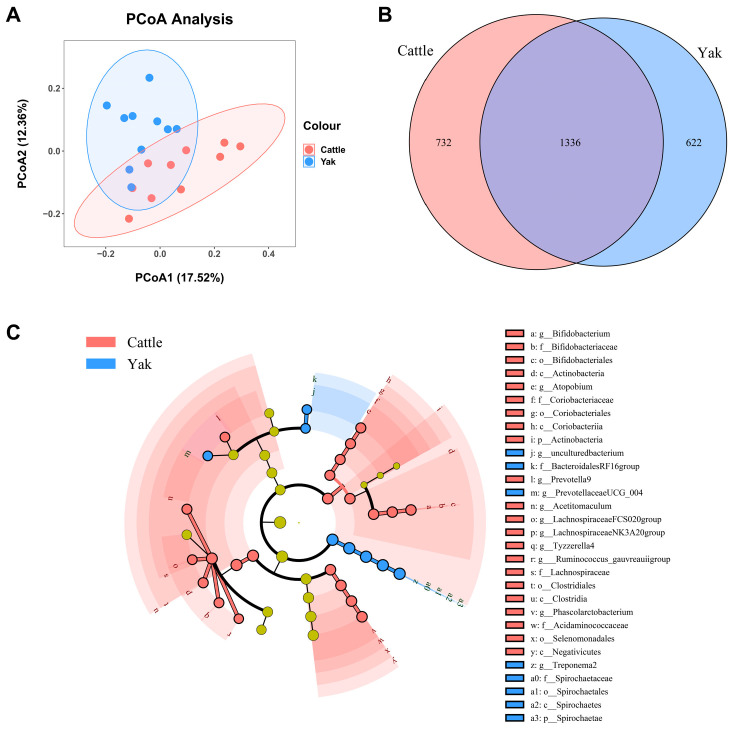
(**A**) PCoA plots and (**B**) Venn diagrams of the fecal bacterial community. (**C**) a cladogram shows fecal bacteria from the phylum level to genus level based on the linear discriminant analysis (LDA) effect size (LEfSe) analysis in yaks and cattle given high concentrate diets.

**Figure 6 microorganisms-11-02399-f006:**
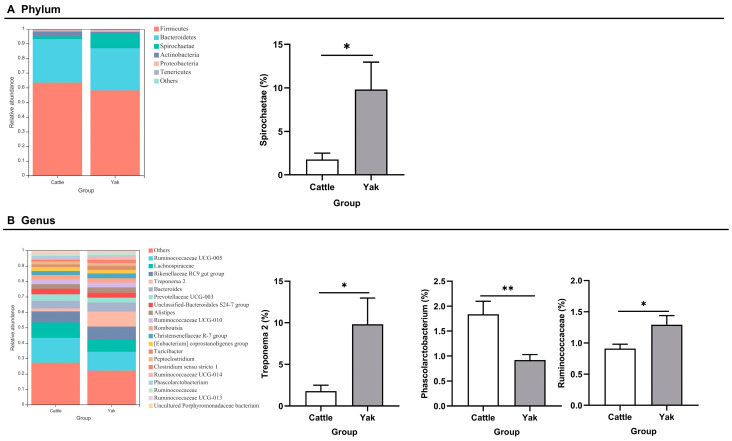
(**A**) Fecal bacterial composition at the phylum level (the relative abundance > 0.5%) and bacteria in cattle and yaks (**B**) Fecal bacterial composition at the genus level (top 20) and bacteria in cattle and yaks given high concentrate diets. A single asterisk (*) indicates a significant difference at *p* < 0.05, while double asterisks (**) indicate a significant difference at *p* < 0.01.

**Figure 7 microorganisms-11-02399-f007:**
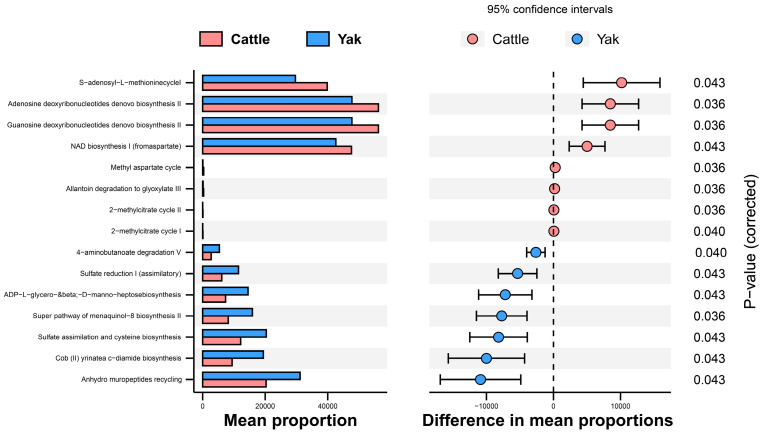
Stamp analysis in the KEGG metabolic pathway of rumen bacteria in yaks and cattle given high concentrate diets.

**Figure 8 microorganisms-11-02399-f008:**
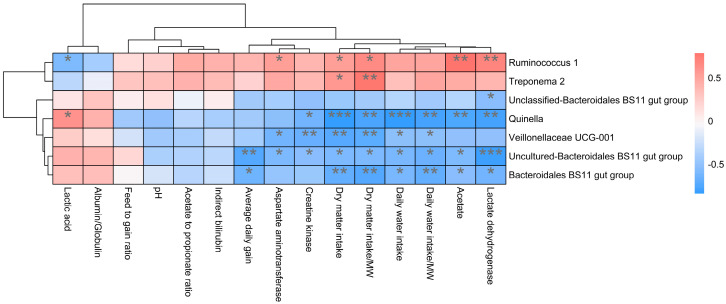
Correlation analysis between rumen differential bacteria and significantly different growth performance, serum biochemical indexes, and rumen fermentation parameters in cattle and yaks given high concentrate diets. A single asterisk (*) indicates a significant difference at *p* < 0.05, double asterisks (**) indicate a significant difference at *p* < 0.01, and triple asterisks (***) indicate a significant difference at *p* < 0.001.

**Table 1 microorganisms-11-02399-t001:** The feed ingredients and chemical composition of diets (DM basis).

Ingredients, %		Nutrient Levels, %	
Oat hay	10	DM	60.89
Corn silage	20	Calcium	0.49
Corn kernels	40	Phosphorus	0.58
Bran	17	Ether extract	2.77
Soybean meal	5	Crude protein	16.19
Rapeseed cake	5	^2^ ME, MJ/kg	11.96
Mineral-vitamin premix ^1^	2	NDF	29.44
Baking soda	1	ADF	16.54

^1^ The premix provided the following per kilogram of diet: vitamin A, 150 kIU/kg; vitamin D3, 30 kIU/kg; vitamin E, 600 IU/kg; vitamin K3, 10 mg/kg; vitamin B1, 10 mg/kg; vitamin B2, 28 mg/kg; vitamin B12, 100 μg/kg; vitamin B9, 4 mg/kg; vitamin B3, 200 mg/kg; vitamin B5, 50 mg/kg; vitamin B7, 2 mg/kg; NaCl, 225 g/kg; I, 16.5 mg/kg; Se, 6 mg/kg; Co, 4 mg/kg; Cu, 0.2 g/kg; Fe, 2.25 g/kg; P, 21 g/kg; Ca, 130 g/kg; Zn, 1.1 g/kg; Mg, 15.5 g/kg; and Mn, 1.6 g/kg. ^2^ ME was calculated value (Liu, 2018), while the other values were measured.

**Table 2 microorganisms-11-02399-t002:** Dry matter intake, water intake, average daily gain, and feed to gain ratio in yaks and cattle given high concentrate diets.

Items	Cattle	Yak	*p*-Value
Initial body weight (kg)	256.9 ± 8.6	194.3 ± 13.7	0.001
Dry matter intake (kg/d)	9.63 ± 0.51	5.49 ± 0.32	<0.001
Daily water intake (kg/d)	35.59 ± 1.07	20.38 ± 0.96	<0.001
Dry matter intake/MW (g/kgW0.75)	128.11 ± 5.52	92.24 ± 3.76	<0.001
Daily water intake/MW (g/kgW0.75)	475.97 ± 19.13	343.96 ± 14.50	<0.001
Average daily gain (kg/d)	1.09 ± 0.08	0.87 ± 0.05	0.039
Feed to gain ratio	9.09 ± 0.60	6.41 ± 0.49	0.004

**Table 3 microorganisms-11-02399-t003:** The serum biochemical indexes in cattle and yaks given high concentrate diets.

Items	Cattle	Yak	*p*-Value
Aspartate aminotransferase (U/L)	91.80 ± 7.62	64.40 ± 3.94	0.005
Alanine aminotransferase (U/L)	27.20 ± 1.97	30.50 ± 2.07	0.263
Total protein (g/L)	76.55 ± 2.48	73.71 ± 3.17	0.491
Albumin (g/L)	35.50 ± 0.48	37.24 ± 0.91	0.113
Globulin (g/L)	41.07 ± 2.70	36.48 ± 2.67	0.242
Albumin/Globulin	0.89 ± 0.06	1.07 ± 0.06	0.048
Alkaline phosphatase (U/L)	218.50 ± 22.93	165.60 ± 13.54	0.066
Total bilirubin (μmol/L)	1.98 ± 0.23	1.23 ± 0.33	0.082
Direct bilirubin (μmol/L)	0.53 ± 0.09	0.50 ± 0.22	0.904
Indirect bilirubin (μmol/L)	1.45 ± 0.15	0.81 ± 0.20	0.021
Creatine kinase (U/L)	477.80 ± 170.74	113.00 ± 12.53	0.047
Lactate dehydrogenase (IU/L)	1431.60 ± 55.05	940.00 ± 53.21	<0.001
Glucose (mmol/L)	4.43 ± 0.12	4.58 ± 0.24	0.578
Urea (mmol/L)	2.95 ± 0.33	4.29 ± 0.35	0.012
Uric acid (μmol/L)	16.40 ± 1.83	34.40 ± 3.06	<0.001
Calcium (mmol/L)	2.71 ± 0.06	2.53 ± 0.10	0.144
Phosphorus (mmol/L)	2.01 ± 0.13	2.24 ± 0.12	0.219
Total cholesterol (mmol/L)	2.81 ± 0.16	1.90 ± 0.18	0.001
Triglyceride (mmol/L)	0.26 ± 0.02	0.34 ± 0.04	0.101

**Table 4 microorganisms-11-02399-t004:** The ruminal parameters in cattle and yaks given high concentrate diets.

Items	Cattle	Yak	*p*-Value
pH	6.13 ± 0.03	6.00 ± 0.03	0.006
Lactate (mmol/L)	0.25 ± 0.04	0.38 ± 0.04	0.028
TVFA (mmol/L)	76.02 ± 7.70	85.11 ± 6.88	0.390
Acetate (%)	59.34 ± 0.81	56.04 ± 0.50	0.003
Propionate (%)	19.48 ± 0.79	21.13 ± 0.65	0.125
Butyrate (%)	15.81 ± 0.87	17.14 ± 0.65	0.232
Isobutyrate (%)	1.56 ± 0.20	1.51 ± 0.19	0.861
Isovalerate (%)	2.02 ± 0.26	2.13 ± 0.26	0.773
Valerate (%)	1.80 ± 0.21	2.06 ± 0.24	0.428
Acetate:Propionate	3.10 ± 0.15	2.67 ± 0.07	0.022

## Data Availability

The raw sequence data were subjected to the Sequence Read Archive (SRA) of the NCBI under accession number PRJNA931567.

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
