# Peer review of "A Comparative Study of Growth Performance, Blood Biochemistry, Rumen Fermentation, and Ruminal and Fecal Bacterial Structure between Yaks and Cattle Raised under High Concentrate Feeding Conditions"

_microorganisms, 2023, doi:10.3390/microorganisms11102399_

Round 1
Reviewer 1 Report (Previous Reviewer 2)
The authors highly improved their paper addressing the reviewer suggestions.
Author Response
Response: Thank you very much for your high evaluation, which is of great significance for our future work and research. In addition, we also appreciate your valuable feedback on the improvement of the article, which has further improved its level.
Reviewer 2 Report (New Reviewer)
Attached

English editing is recommended
Author Response
Point 1: INTRODUCTION: The section should be amended using more recent scientific references.
Response 1: Thank you very much for your valuable feedback. We have updated the references in this section. The revised content is as follows:
2.Huang, X.; Mi, J.; Denman, S.E.; Basangwangdui; Pingcuozhandui; Zhang, Q.; Long, R.; McSweeney, C.S. Changes in rumen microbial community composition in yak in response to seasonal variations. J Appl Microbiol 2022, 132, 1652-1665, doi: 10.1111/jam.15322.
- Zhao, C.; Wang, L.; Ke, S.; Chen, X.; Kenéz, Á.; Xu, W.; Wang, D.; Zhang, F.; Li, Y.; Cui, Z.; et al. Yak rumen microbiome elevates fiber degradation ability and alters rumen fermentation pattern to increase feed efficiency. Anim Nutr 2022, 11, 201-214, doi: 10.1016/j.aninu.2022.07.014.
- Liu, H.; Ran, T.; Zhang, C.; Yang, W.; Wu, X.; Degen, A.; Long, R.; Shi, Z.; Zhou, J. Comparison of rumen bacterial communities between yaks (Bos grunniens) and Qaidam cattle (Bos taurus) fed a low protein diet with different energy levels. Front Microbiol 2022, 13, 982338, doi: 10.3389/fmicb.2022.982338.
- Nagata, R.; Kim, Y. H.; Ohkubo, A.; Kushibiki, S.; Ichijo, T.; Sato, S. Effects of repeated subacute ruminal acidosis challenges on the adaptation of the rumen bacterial community in Holstein bulls. J Dairy Sci 2018, 101, 4424–4436, doi: 10.3168/jds.2017-13859.
- Ogata, T.; Makino, H.; Ishizuka, N.; Iwamoto, E.; Masaki, T.; Ikuta, K.; Kim, Y.H.; Sato, S. Long-term high-grain diet altered the ruminal pH, fermentation, and composition and functions of the rumen bacterial community, leading to enhanced lactic acid production in Japanese black beef cattle during fattening. PLoS One 2019, 14, e0225448, doi: 10.1371/JOURNAL.PONE.0225448.
Point 2: INTRODUCTION: Your hypothesis of the experiment should be included.
Response 2: Thanks for your valuable comments. We have added the hypothesis of the experiment in the manuscript. The supplemented content is as follows:
Therefore, in the current study, we hypothesized that there is a difference in the metabolic ability of cattle and yaks to high-concentrate diets, including the differences in growth performance, blood biochemistry, rumen fermentation, and ruminal and fecal bacterial structure between yaks and cattle.
Point 3: CONCLUSION: The section seems well, however, can be shortened.
Response 3: Thanks for your valuable comments. We have shortened the conclusion in the manuscript. The revised content is as follows:
In general, yaks and cattle have been mainly grazed and captive breeding for a long time, respectively. When feeding the high-concentrate diet, they showed a difference in growth performance, blood biochemical indices, rumen fermentation parameters, and rumen microbiota. In addition, the fecal bacterial structures of cattle and yaks are also different, suggesting that yaks may have stronger digestive capacity in the hindgut.
This manuscript is a resubmission of an earlier submission. The following is a list of the peer review reports and author responses from that submission.
Round 1
Reviewer 1 Report
1. The objective of the study is not new. There is no clear hypothesis to compare the two species.
2. The requirements of the two species are different, for this reason the same diet cannot be evaluated.
3. The ration is poorly formulated, the phosphorus level is higher than calcium, from this imbalance urolithiasis can be induced and change the hematic variables. Nor is it necessary to add group B vitamins, there is no reason.
4. The authors mention that yaks had better growth performance and it is not true. The authors only evaluated the weight gain and the differences in initial body weight between the two species is different.
5. I am recommended to study the nutritional requirements of yaks, digestibility, net energy efficiency, weight gain, and carcass and meat quality. Using a single diet to compare both species is not substantiated.
Author Response
Response to Reviewer 1 Comments
Point 1: The objective of the study is not new. There is no clear hypothesis to compare the two species.
Response 1: Thank you very much for the comments. Most of the research on nutrient metabolism in yaks and cattle has been focused on grazing or roughage conditions, and there have been few studies reporting their reactions to high-concentrate diets. Based on this, this article has conducted relevant research. At the same time, we have supplemented the experimental hypotheses as follows:
However, there is less information on the metabolic capacity, rumen microbiota, and health status of yaks comparing cattle given high-concentrate diets. Therefore, in the current study, we hypothesized that there is a difference in the metabolic ability of cattle and yaks to high-concentrate diets, including the differences in growth performance, blood biochemistry, rumen fermentation, and ruminal and fecal bacterial structure between yaks and cattle. To achieve this goal, rumen fluid, serum, and fecal samples from yak and cattle after two months of the feeding period to investigate differences in relevant indicators.
Point 2: The requirements of the two species are different, for this reason the same diet cannot be evaluated.
Response 2: Thank you very much for your question, which is of great significance for our future related work. Firstly, species and diet are very important factors that affect the metabolism and utilization of nutrients. Therefore, when studying the differences between the two, we ensured the same diet. It is precisely because of the differences in nutritional requirements between the two that lead to differences in energy metabolism and utilization in high-concentrate diets.
Point 3: The ration is poorly formulated, the phosphorus level is higher than calcium, from this imbalance urolithiasis can be induced and change the hematic variables. Nor is it necessary to add group B vitamins, there is no reason.
Response 3: We apologize for neglecting the impact of trace elements when designing our diet. We also appreciate your suggestions and will make improvements in our future work.
Point 4: The authors mention that yaks had better growth performance and it is not true. The authors only evaluated the weight gain and the differences in initial body weight between the two species is different.
Response 4: Thank you very much for pointing out the issue of growth performance. Firstly, due to the short stature formed by yaks during long-term evolution, it is difficult to find cattle and yaks with the same age and weight. Therefore, under the same age conditions, the smaller weight of yaks is an inherent difference between the two. Secondly, while measuring the average daily gain, we also measured the feed-to-gain ratio, and the results showed that yaks had a significantly lower feed to gain ratio. Therefore, we speculate that under the same dietary conditions, yaks can more effectively utilize dietary nutrients compared to cattle.
Point 5: I am recommended to study the nutritional requirements of yaks, digestibility, net energy efficiency, weight gain, and carcass and meat quality. Using a single diet to compare both species is not substantiated.
Response 5: Thank you very much for your question, which is of great significance for our future related work. We will focus more on the nutritional needs, nutrient digestibility, and meat quality of yaks.
Reviewer 2 Report
See attached file

Author Response
Response to Reviewer 2 Comments
Point 1: please discuss also the ruminal parameters. Indeed, even if there were not significant
differencs between yak and cattle, I suggest to add some sentences, may be comparing the
results with those obtained in other animal species (buffalo, for istance, as the authors did for
serum parameters).
Response 1: Thank you very much for your valuable feedback. We have supplemented the content of this part. The supplemented content is as follows:
The pH and the concentrations of VFAs are the main indicators of the status of rumen fermentation. In the present study, the ratio of acetate to propionate ranged between 3.10 and 2.67. A ratio greater than 2.2 was reported to have a positive effect on the rumen and productive performance of cattle, which would indicate that all the yaks and cattle had a well-functioning rumen. The acetate was significantly higher in cattle than in yaks in the study, which is linked closely to fiber degradation. This shows that cattle are more effective in utilizing fiber in high-concentrate diets compared to yaks. In addition, the lactate content was significantly higher in yaks than in cattle, this may also indicate that yaks can more effectively utilize starch in high-concentrate diets, which further reduces the pH value in the rumen of yaks. It may be an important reason why the rumen pH of yaks is lower than that of cattle.
Point 2: line 172: please add “accordingly, also buffalo shows lower daiy weight gain than bovine
(Infascelli et al., 2004)”.
Response 2: Thanks for your valuable comments. We have added the referee in the manuscript.
Point 3: line 194: please add “Anyway the serum urea level in yak was also higher than that found in
buffalo (Lombardi et al., 2017)”.
Response 3: The suggestion is helpful for our writing and we have supplemented the suggestion.
Point 4: line 206: as a reference for buffalo cholesterol please add also the above mentioned Lombardi et al. (2017) which found the same level.
Response 4: Thanks for your significant suggestion, we have supplemented it in the manuscript.
Point 5: I suggest to rewrite the conclusions which in present form appear a repetition of the results. Instead I should emphatize the importance of this trial for its contribution to increase the knowledge of yak physiology which will allow to better bred this animal.
Response 5: Thank you very much for your comment, we have rewritten the conclusions and supplemented the importance of this trial. The revised conclusion is as follows:
In general, yaks and cattle have been mainly grazed and captive breeding for a long time, respectively. When feeding the high-concentrate diet, they showed a differ-ence in growth performance, blood biochemical indices, rumen fermentation parame-ters, and rumen microbiota. In addition, the fecal bacterial structures of cattle and yaks are also different, suggesting that yaks may have stronger digestive capacity in the hindgut. In the future, the functions of microbiota and their response to high concen-trated diet in the gastrointestinal tract of yaks, especially in the hindgut, are needed in-depth study using metagenomics and transcriptomics. In conclusion, under the condi-tions of high concentrated diet, the study is of fundamental significance to understand-ing the growth performance, blood biochemistry, rumen fermentation, ruminal and fe-cal bacterial structure between yaks and cattle, which provides a theoretical basis for the fattening method and precision feeding of yaks, so as to maximize the feed effi-ciency on the basis of ensuring the health of yaks, then it can alleviate the pressure of grassland ecosystem and reduce environmental pollution.

Round 2
Reviewer 1 Report
The first review indicates that the article has problems in the objective and methodology, for which my opinion was not approved. The authors make slight corrections that do not convince me. I ask the editor to request another reviewer's opinion, I do not agree with its publication